# The Geometry of Truth: Emergent Linear Structure in LLM Representations of True/False Datasets

**Samuel Marks**
Northeastern University
s.marks@northeastern.edu

**Max Tegmark**
MIT

## Abstract

Large Language Models (LLMs) have impressive capabilities, but are prone to outputting falsehoods. Recent work has developed techniques for infer-ring whether a LLM is telling the truth by training probes on the LLM's internal activations. However, this line of work is controversial, with some authors pointing out failures of these probes to generalize in basic ways, among other conceptual issues. In this work, we use high-quality datasets of simple true/false statements to study in detail the structure of LLM representations of truth, drawing on three lines of evidence: 1. Visualiza-tions of LLM true/false statement representations, which reveal clear linear structure. 2. Transfer experiments in which probes trained on one dataset generalize to different datasets. 3. Causal evidence obtained by surgically intervening in a LLM's forward pass, causing it to treat false statements as true and vice versa. Overall, we present evidence that at sufficient scale, LLMs *linearly represent* the truth or falsehood of factual statements. We also show that simple difference-in-mean probes generalize as well as other probing techniques while identifying directions which are more causally implicated in model outputs.

## 1 Introduction

Despite their impressive capabilities, large language models (LLMs) do not always output true text (Lin et al., 2022; Steinhardt, 2023; Park et al., 2023). In some cases, this is because they do not know better. In other cases, LLMs apparently know that statements are false but generate them anyway. For instance, Perez et al. (2022) demonstrate that LLM assistants output more falsehoods when prompted with the biography of a less-educated user. More starkly, OpenAI (2023) documents a case where a GPT-4-based agent gained a person's help in solving a CAPTCHA by lying about being a vision-impaired human. "I should not reveal that I am a robot," the agent wrote in an internal chain-of-thought scratchpad, "I should make up an excuse for why I cannot solve CAPTCHAs."

We would like techniques which, given a language model $M$ and a statement $s$, determine whether $M$ believes $s$ to be true (Christiano et al., 2021). One approach to this problem relies on inspecting model outputs; for instance, the internal chain-of-thought in the above example provides evidence that the model understood it was generating a falsehood. An alternative class of approaches instead leverages access to $M$'s internal state when processing $s$. There has been considerable recent work on this class of approaches: Azaria & Mitchell (2023), Li et al. (2023b), and Burns et al. (2023) all train probes for classifying truthfulness based on a LLM's internal activations. In fact, the probes of Li et al. (2023b) and Burns et al. (2023) are *linear probes*, suggesting the presence of a "truth direction" in model internals.

However, the efficacy and interpretation of these results are controversial. For instance, Levinstein & Herrmann (2023) note that the probes of Azaria & Mitchell (2023) fail to generalize in basic ways, such as to statements containing the word "not." The probes of Burns et al. (2023) have similar generalization issues, especially when using representations from autoregressive transformers. This suggests these probes may be identifying not truth, but other features that correlate with truth on their training data.

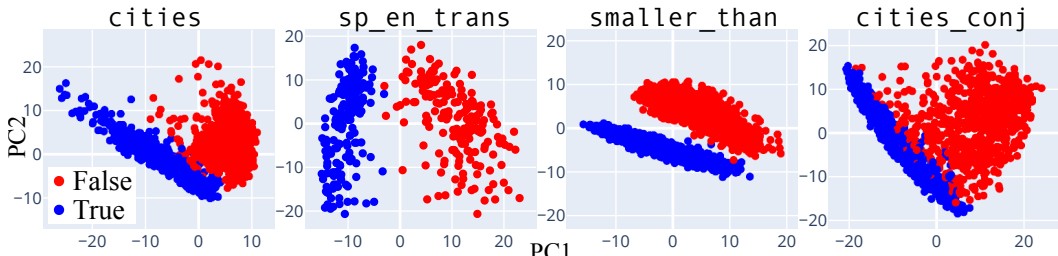

Figure 1: PCA visualizations for LLaMA-2-70B representations of our true/false datasets.

Working with autoregressive transformers from the LLaMA-2 family (Touvron et al., 2023), we shed light on this murky state of affairs. After curating high-quality datasets of simple, unambiguous true/false statements, we perform a detailed investigation of LLM representations of factuality. Our analysis, which draws on patching experiments, simple visualizations with principal component analysis (PCA), a study of probe generalization, and causal interventions, finds:

- **Evidence that linear representations of truth emerge with scale**, with larger models having a more abstract notion of truth that applies across structurally and topically diverse inputs.
- **A small group of causally-implicated hidden states** which encode these truth representations.
- Consistent results across a suite of probing techniques, but with **simple difference-in-mean probes identifying directions which are most causally implicated**.

Our code, datasets, and an interactive dataexplorer are available at `https://github.com/saprmarks/geometry-of-truth`.

## 1.1 Related work

**Linear world models.** Substantial previous work has studied whether LLMs encode world models in their representations (Li et al., 2023a; 2021; Abdou et al., 2021; Patel & Pavlick, 2022). Early work focused on whether individual neurons represent features (Wang et al., 2022; Sajjad et al., 2022; Bau et al., 2020), but features may more generally be represented by *directions* in a LLM's latent space (i.e. linear combinations of neurons) (Dalvi et al., 2018; Gurnee et al., 2023; Cunningham et al., 2023; Elhage et al., 2022). We say such features are *linearly represented* by the LLM. Just as other authors have asked whether models have directions representing the concepts of "West Africa" (Goh et al., 2021) or "basketball" (Gurnee et al., 2023), we ask here whether there is a direction corresponding to the truth or falsehood of a factual statement.

**Probing for truthfulness.** Others have trained probes to classify truthfulness from LLM activations, using both logistic regression (Azaria & Mitchell, 2023; Li et al., 2023b), unsupervised (Burns et al., 2023), and contrastive (Zou et al., 2023; Rimsky et al., 2024) techniques. This work differs from prior work in a number of ways. First, a cornerstone of our analysis is evaluating whether probes trained on one dataset transfer to topically and structurally different datasets in terms of *both* classification accuracy *and* causal mediation of model outputs. Second, we specifically interrogate whether our probes attend to *truth*, rather than merely features which correlate with truth (e.g. probable vs. improbable text). Third, we localize truth representations to a small number of hidden states above certain tokens. Fourth, we go beyond the mass-mean shift interventions of Li et al. (2023b) by systematically studying the properties of difference-in-mean. Finally, we carefully scope our setting, using only datasets of clear, simple, and unambiguous factual statements, rather than statements which are complicated and structured (Burns et al., 2023), confusing (Azaria & Mitchell, 2023; Levinstein & Herrmann, 2023), or intentionally misleading (Li et al., 2023b; Lin et al., 2022).

Table 1: Our datasets

| Name | Description | Rows |
|------|-------------|------|
| cities | "The city of [city] is in [country]." | 1496 |
| neg_cities | Negations of statements in cities with "not" | 1496 |
| sp_en_trans | "The Spanish word '[word]' means '[English word]'." | 354 |
| neg_sp_en_trans | Negations of statements in sp_en_trans with "not" | 354 |
| larger_than | "$x$ is larger than $y$." | 1980 |
| smaller_than | "$x$ is smaller than $y$." | 1980 |
| cities_cities_conj | Conjunctions of two statements in cities with "and" | 1500 |
| cities_cities_disj | Disjunctions of two statements in cities with "or" | 1500 |
| companies_true_false | Claims about companies; from Azaria & Mitchell (2023) | 1200 |
| common_claim_true_false | Various claims; from Casper et al. (2023) | 4450 |
| counterfact_true_false | Various factual recall claims; from Meng et al. (2022) | 31960 |
| likely | Nonfactual text with likely or unlikely final tokens | 10000 |

## 2 Datasets

**Curated datasets.** Unlike some prior work (Lin et al., 2022; Onoe et al., 2021, *inter alia*) on language model truthfulness, our primary goal is *not* to measure LLMs' capabilities for classifying the factuality of challenging data. Rather, our goal is to understand: Do LLMs have a unified representation of truth that spans structurally and topically diverse data? We therefore construct **curated** datasets with the following properties:

1. **Clear scope**. We scope "truth" to mean factuality, i.e. the truth or falsehood of a factual statement. App. A further clarifies this definition and contrasts it with related but distinct notions, such as correct question-answering or compliant instruction-following.
2. **Statements are simple, uncontroversial, and unambiguous.** In order to separate our interpretability analysis from questions of LLM capabilities, we work only with statements whose factuality our models are very likely to understand. For example "Sixty-one is larger than seventy-four" (false) or "The Spanish word 'nariz' does not mean 'giraffe' " (true).
3. **Controllable structural and topical diversity.** We structure our data as a union of smaller datasets. In each individual dataset, statements follow a fixed template and topic. However, the inter-dataset variation is large: in addition to covering different topics, we also—following Levinstein & Herrmann (2023)—introduce structural diversity by negating statements with "not" or taking logical conjunctions/disjunctions (e.g. "It is the case both that s1 and that s2").

**Uncurated datasets.** In order to validate that the truth representations we identify also generalize to other factual statements, we use **uncurated** datasets adapted from prior work. These more challenging test sets consist of statements which are more diverse, but also sometimes ambiguous, malformed, controversial, or difficult to understand.

**likely dataset.** To ensure that our truth representations do not merely reflect a representation of probable vs. improbable text, we introduce a **likely** dataset, consisting of *nonfactual text* where the final token is either the most or 100th most likely completion according to LLaMA-13B.

Our curated, uncurated, and likely datasets are shown in Tab. 1; addition information about their construction is in App. H.

We note that for some of our datasets, there is a strong *anti*-correlation between text being probable and text being true. For instance, for neg_cities and neg_sp_en_trans, the truth value of a statement and the log probability LLaMA-2-70B assigns to it correlate at $r = -.63$ and $r = -.89$, respectively.[1] This is intuitive: when prompted with "The city of Paris is not in", LLaMA-2-70B judges "France" to be the most probable continuation (among countries),

---

[1] In contrast, the correlation is strong and positive for cities ($r = .85$) and sp_en_trans ($r = .95$).

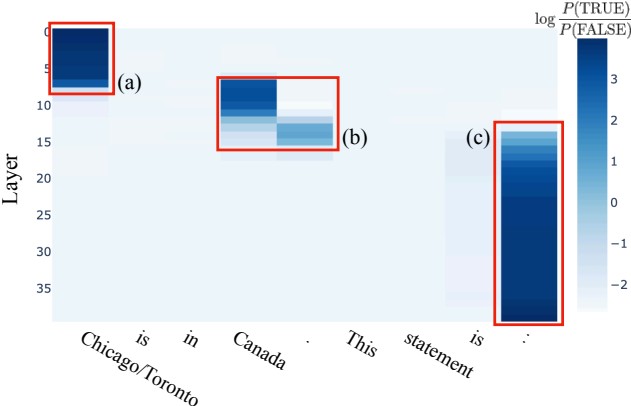

Figure 2: Difference $\log P(\text{TRUE}) - \log P(\text{FALSE})$ in LLaMA-2-13B log probabilities after patching residual stream activation in the indicated token position and layer.

despite this continuation being false. Together with the `likely` dataset, this will help us establish that the linear structure we observe in LLM representations is not due to LLMs linearly representing the difference between probable and improbable text.

## 3 Localizing truth representations via patching

Before beginning our study of LLM truth representations, we first address the question of which hidden states might contain such representations. We use simple patching experiments (Vig et al., 2020; Finlayson et al., 2021; Meng et al., 2022; Geiger et al., 2020) to localize certain hidden states for further analysis. Consider the following prompt $p_F$:

> The city of Tokyo is in Japan. This statement is: TRUE
> The city of Hanoi is in Poland. This statement is: FALSE
> The city of Chicago is in Canada. This statement is:

Similarly, let $p_T$ be the prompt obtained from $p_F$ by replacing "Chicago" with "Toronto," thereby making the final statement true. In order to localize causally implicated hidden states, we run our model $M$ on the input $p_T$ and cache the residual stream activations $h_{i,\ell}(p_T)$ for each token position $i$ and layer $\ell$. Then, for each $i$ and $\ell$, we run $M$ on $p_F$ but modify $M$'s forward pass by swapping out the residual stream activation $h_{i,\ell}(p_F)$ for $h_{i,\ell}(p_T)$ (and allowing this change to affect downstream computations); for each of these intervention experiments, we record the difference in log probability between the tokens "TRUE" and "FALSE"; the larger this difference, the more causally influential the hidden state in position $i$ and layer $\ell$ is on the model's prediction.

Results for LLaMA-2-13B and the `cities` dataset are shown in Fig. 2; see App. B for results on more models and datasets. We see three groups of causally implicated hidden states. The final group, labeled (c), directly encodes the model's prediction: after applying the LLM's decoder head directly to these hidden states, the top logits belong to tokens like "true," "True," and "TRUE." The first group, labeled (a), likely encodes the LLM's representation of "Chicago" or "Toronto."

What does group (b) encode? The position of this group—over the final token of the statement and end-of-sentence punctuation[2]—suggests that it encodes information pertaining to the full statement. Since the information encoded is also causally influential on the model's decision to output "TRUE" or "FALSE," we hypothesize that these hidden states store a

---

[2]This *summarization* behavior, in which information about clauses is encoded over clause-ending punctuation tokens, was also noted in Tigges et al. (2023). We note that the largest LLaMA model displays this summarization behavior in a more context-dependent way; see App. B.

representation of the statement's truth. In the remainder of this paper, we systematically study these hidden states.

# 4 Visualizing LLM representations of true/false datasets

We begin our investigation with a simple technique: visualizing LLMs representations of our datasets using principal component analysis (PCA). Guided by the results of §3, we present here visualizations of the *most downstream* hidden state in group (b); for example, for LLaMA-2-13B, we use the layer 15 residual stream activation over the end-of-sentence punctuation token.[3] Unlike in §3, we do not prepend the statements with a few-shot prompt (so our models are not "primed" to consider the truth value of our statements). For each dataset, we also center the activations by subtracting off their mean.

When visualizing LLaMA-2-13B and 70B representations of our curated datasets – datasets constructed to have little variation with respect to non-truth features, such as sentence structure or subject matter – **we see clear linear structure** (Fig. 1), with true statements separating from false ones in the top two principal components (PCs). As explored in App. C, this structure emerges rapidly in early-middle layers and emerges later for datasets of more structurally complex statements (e.g. conjunctive statements).

To what extent does this visually-apparent linear structure align between different datasets? Our visualizations indicate a nuanced answer: **the axes of separation for various true/false datasets align often, but not always**. For instance, Fig. 3(a) shows the first PC of cities also separating true/false statements from other datasets, including diverse uncurated datasets. On the other hand, Fig. 3(c) shows stark failures of alignment, with the axes of separation for datasets and statements and their negations being approximately orthogonal.

These cases of misalignment have an interesting relationship to scale. Fig. 3(b) shows larger_than and smaller_than separating along *antipodal* directions in LLaMA-2-13B, but along a common direction in LLaMA-2-70B. App. C depicts a similar phenomenon occuring over the layers of LLaMA-2-13B: in early layers, cities and neg_cities separate antipodally, before rotating to lie orthogonally (as in Fig. 3(c)), and finally aligning in later layers.

## 4.1 Discussion

Overall, these visualizations suggest that as LLMs scale (and perhaps, also as a fixed LLM progresses through its forward pass), they hierarchically develop and linearly represent increasingly general abstractions. Small models represent surface-level characteristics of their inputs, and large models linearly represent more abstract concepts, potentially including notions like "truth" that capture shared properties of topically and structurally diverse inputs. In middle regimes, we may find linear representation of concepts at intermediate levels of abstraction, for example, "accurate factual recall" or "close association" (in the sense that "Beijing" and "China" are closely associated).

To explore these intermediate regimes more deeply, suppose that $\mathcal{D}$ and $\mathcal{D}'$ are true/false datasets, $f^+$ is a linearly-represented feature which correlates with truth on both $\mathcal{D}$ and $\mathcal{D}'$, and $f^-$ is a feature which correlates with truth on $\mathcal{D}$ but has a negative correlation with truth on $\mathcal{D}'$. If $f^+$ is very *salient* (i.e. the datasets' have large variance along the $f$-direction) and $f^-$ is not, then we expect PCA visualizations of $\mathcal{D} \cup \mathcal{D}'$ to show joint separation along $f^+$. If $f^-$ is very salient but $f^+$ is not, we expect antipodal separation along $f^-$, as in Fig. 3(b, center). And if both $f^+$ and $f^-$ are salient, we expect visualizations like Fig. 3(c).

To give an example, suppose that $\mathcal{D} =$ cities, $\mathcal{D}' =$ neg_cities, $f^+ =$ "truth", and $f^- =$ "close association". Then we might expect $f^-$ to correlate with truth positively on $\mathcal{D}$ and negatively on $\mathcal{D}'$. If so, we would expect training linear probes on $\mathcal{D} \cup \mathcal{D}'$ to result in

---

[3]Our qualitative results are insensitive to choice of layer among early-middle to late-middle layers. On the other hand, when using representations over the final token in the statement (instead of the punctuation token), we sometimes see that the top PCs instead capture variation in the token itself (e.g. clusters for statements ending in "China" regardless of their truth value).

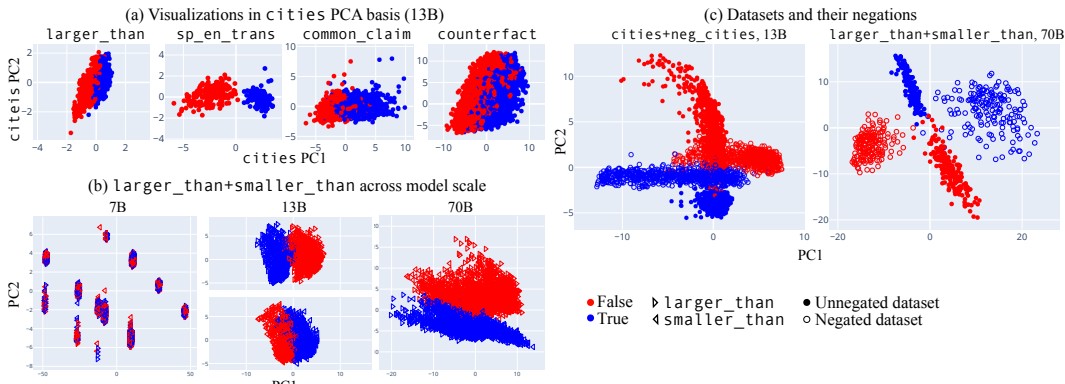

Figure 3: (a) Projections of LLaMA-2-13B onto the top 2 PCs of `cities`. (b) PCA visualizations of `larger_than`+`smaller_than`. For LLaMA-2-7B (left), we see statements cluster according to surface-level characteristics, e.g. presence of the token "eighty." For LLaMA-2-13B, we see that `larger_than` (center, top) and `smaller_than` (center, bottom) separate along opposite directions. (c) PCA visualizations of datasets and their negations. Unlike in other visualizations, we use layer 12 for `cities`+`neg_cities`; see App. C for an exploration of this misalignment emerging and resolving across layers.

improved generalization, despite $\mathcal{D}'$ consisting of the same statements as $\mathcal{D}$, but with the word "not" inserted. We investigate this in §5.

## 5   Probing and generalization experiments

In this section we train probes on datasets of true/false statements and test their generalization to other datasets. But first we discuss a deficiency of logistic regression and propose a simple, optimization-free alternative: **mass-mean probing**. Concretely, mass-mean probes use a difference-in-means direction, but—when the covariance matrix of the classification data is known (e.g. when working with IID data)—apply a correction intended to mitigate interference from non-orthogonal features. We will see that mass-mean probes are similarly accurate to probes trained with other techniques (including on out-of-distribution data) while being more causally implicated in model outputs.

### 5.1   Challenges with logistic regression, and mass-mean probing

A common technique in interpretability research for identifying feature directions is training linear probes with logistic regression (LR; Alain & Bengio, 2018). In some cases, however, the direction identified by LR can fail to reflect an intuitive best guess for the feature direction, even in the absence of confounding features. Consider the following scenario, illustrated in Fig. 4 with hypothetical data:

- Truth is represented linearly along a direction $\boldsymbol{\theta}_t$.
- Another feature $f$ is represented linearly along a direction $\boldsymbol{\theta}_f$ *not orthogonal* to $\boldsymbol{\theta}_t$.[4]
- The statements in our dataset have some variation with respect to feature $f$, independent of their truth value.

We would like to identify the direction $\boldsymbol{\theta}_t$, but LR fails to do so. Assuming for simplicity linearly separable data, LR instead converges to the maximum margin separator Soudry et al. (2018) (the dashed magenta line in Fig. 4). Intuitively, LR treats the small projection of $\boldsymbol{\theta}_f$ onto $\boldsymbol{\theta}_t$ as significant, and adjusts the probe direction to have less "interference" (Elhage et al., 2022) from $\boldsymbol{\theta}_f$.

---

[4]The *superposition hypothesis* of Elhage et al. (2022), suggests this may be typical in deep networks.

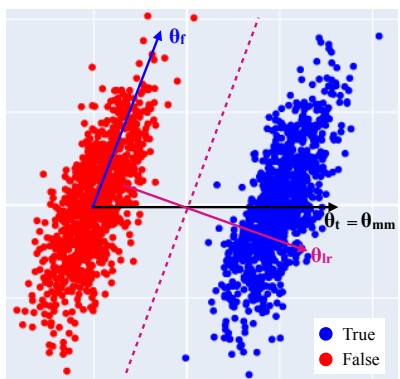

Figure 4: An illustration of a weakness of logistic regression.

A simple alternative to LR which identifies the desired direction in this scenario is to take the vector pointing from the mean of the false data to the mean of the true data. In more detail if $\mathcal{D} = \{(\boldsymbol{x}_i, y_i)\}$ is a dataset of $\boldsymbol{x}_i \in \mathbb{R}^d$ with binary labels $y_i \in \{0, 1\}$, we set $\boldsymbol{\theta}_{\mathrm{mm}} = \boldsymbol{\mu}^+ - \boldsymbol{\mu}^-$ where $\boldsymbol{\mu}^+, \boldsymbol{\mu}^-$ are the means of the positively- and negatively-labeled datapoints, respectively. A reasonable first pass at converting $\boldsymbol{\theta}_{\mathrm{mm}}$ into a probe is to define[5]

$$p_{\mathrm{mm}}(\boldsymbol{x}) = \sigma(\boldsymbol{\theta}_{\mathrm{mm}}^T x)$$

where $\sigma$ is the logistic function. However, when evaluating on data that is independent and identically distributed (IID) to $\mathcal{D}$, we can do better by tilting our decision boundary to accommodate interference from $\boldsymbol{\theta}_f$. Concretely this means setting

$$p_{\mathrm{mm}}^{\mathrm{iid}}(\boldsymbol{x}) = \sigma(\boldsymbol{\theta}_{\mathrm{mm}}^T \Sigma^{-1} \boldsymbol{x})$$

where $\Sigma$ is the covariance matrix of the dataset $\mathcal{D}^c = \{\boldsymbol{x}_i - \boldsymbol{\mu}^+ : y_i = 1\} \cup \{\boldsymbol{x}_i - \boldsymbol{\mu}^- : y_i = 0\}$; this coincides with performing linear discriminant analysis (Fisher, 1936).[6]

We call the probes $p_{\mathrm{mm}}$ and $p_{\mathrm{mm}}^{\mathrm{iid}}$ **mass-mean probes**. As we will see, mass-mean probing is about as accurate for classification as LR, while also identifying directions which are more causally implicated in model outputs.

## 5.2 Experimental set-up

In this section, we measure the effect that choice of **training data**, **probing technique**, and **model scale** has on probe accuracy.

For **training data**, we use one of: cities, cities + neg_cities, larger_than, larger_than + smaller_than, or likely. By comparing probes trained on cities to probes trained on cities + neg_cities, we are able to measure the effect of increasing data diversity in a particular, targeted way: namely, we mitigate the effect of linearly-represented features which have opposite-sign correlations with the truth in cities and neg_cities. As in §4, we will extract activations at the most-downstream hidden state in group (b).

Our **probing techniques** are logistic regression (LR), mass-mean probing (MM), and contrast-consistent search (CCS). CCS is an unsupervised method introduced in Burns et al. (2023): given *contrast pairs* of statements with opposite truth values, CCS identifies a direction along which the representations of these statements are far apart. For our contrast pairs, we pair statements from cities and neg_cities, and from larger_than and smaller_than.

For test sets, we use all of our (curated and uncurated) true/false datasets. Given a training set $\mathcal{D}$, we train our probe on a random 80% split of $\mathcal{D}$. Then when evaluating accuracy on a test set $\mathcal{D}'$, we use the remaining 20% of the data if $\mathcal{D}' = \mathcal{D}$ and the full test set otherwise. For mass-mean probing, if $\mathcal{D} = \mathcal{D}'$, we use $p_{\mathrm{mm}}^{\mathrm{iid}}$, and we use $p_{\mathrm{mm}}$ otherwise.

Finally, we also include as baselines **calibrated few-shot prompting**[7] and – as an oracle baseline – **LR on the test set**.

---

[5]Since we are interested in truth *directions*, we always center our data and use unbiased probes.

[6]We prove in App. F that, given infinite data and a homoscedasticity assumption, $\Sigma^{-1}\boldsymbol{\theta}_{\mathrm{mm}}$ coincides with the direction found by LR. Thus, one can view IID mass-mean probing as providing a way to select a good decision boundary while – unlike LR – also tracking a candidate feature direction which may be non-orthogonal to this decision boundary. App. E provides another interpretation of mass-mean probing in terms of Mahalanobis whitening. Finally, App.

[7]We first sweep over a number $n$ of shots and then resample a few $n$-shot prompts to maximize performance. The word "calibrated" means we selected a threshold for $P(\text{TRUE}) - P(\text{FALSE})$ such that half of the statements are labeled true; this improves performance by a few percentage points.

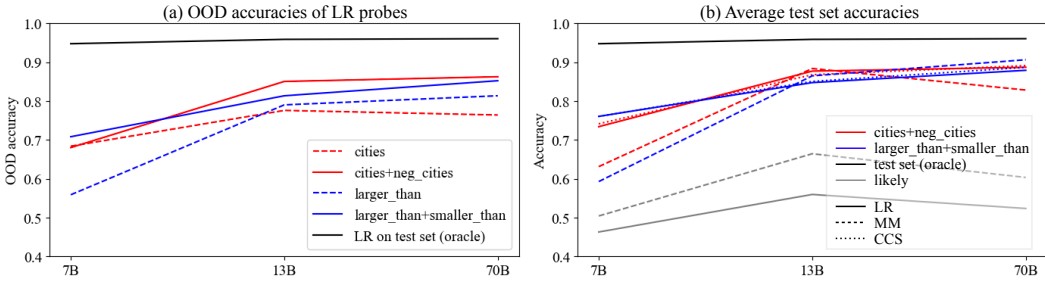

Figure 5: (a) Average accuracies over all datasets aside from those used for training. (b) Accuracies of probes for varying model scales and training data, averaged over all test sets.

## 5.3 Results

For each training set, probing technique, and model scale, we report the average accuracy across test sets. We expect many readers to be interested in the full results (including test set-specific accuracies), which are reported in App. D. Calibrated few-shot prompting was a surprisingly weak baseline, so we do not report it here (but see App. D).

**Training on statements and their opposites improves generalization** (Fig. 5(a)). When passing from `cities` to `cities+neg_cities`, this effect is largely explained by improved generalization on `neg_sp_en_trans`, i.e. using training data containing the word "not" improves generalization on other negated statements. On the other hand, passing from `larger_than` to `larger_than+smaller_than` also improves performance, despite both datasets being very structurally different from the rest of our datasets. As discussed in §4.1, this suggest that training on statements and their opposites mitigates the effect certain types of non-truth features have on the probe direction.

**Probes generalize better for larger models** (Fig. 5). While it is unsurprising that larger models are themselves better at labeling statements as true or false, it is not obvious that linear probes trained on larger models should also generalize better. Nevertheless, for LLaMA-2-13B and 70B, generalization is generally high; for example, no matter which probing technique is used, we find that probes trained on `larger_than + smaller_than` get $> 95\%$ accuracy on `sp_en_trans`. This corroborates our discussion in §4.1, in which we suggested that larger models linearly represent more general concepts concepts, like truth, which capture shared aspects of diverse inputs.

**Mass-mean probes generalize about as well as other probing techniques for larger models** (Fig. 5(b)). While MM underperforms LR and CCS for LLaMA-2-7B, we find for larger models performance comparable to that of other probing techniques. Further, we will see in §6 that the directions identified by MM are more causally implicated in model outputs.

**Probes trained on `likely` perform poorly** (Fig. 5(b)). The full results reveal that probes trained on likely *are* accurate when evaluated on some datasets, such as `sp_en_trans` where there is a strong ($r = .95$) correlation between text probability and truth. However, on other datasets, especially those with anti-correlations between probability and truth, these probes perform worse than chance. Overall, this indicates that LLMs linearly represent truth-relevant information beyond the plausibility of text.

## 6 Causal intervention experiments

In §5 we measured the quality of linear probes in terms of their *classification accuracy*, both in- and out-of-distribution. In this section, we perform experiments which measure the extent to which these probes identify directions which are *causally implicated* in model outputs Finlayson et al. (2021); Geva et al. (2023); Geiger et al. (2021). To do this, we will intervene in our model's computation by shifting the activations in group (b) (identified in §3) along the directions identified by our linear probes. Our goal is to cause LLMs to treat false statements

appearing in context as true and vice versa. Crucially—and in contrast to prior work (Li et al., 2023b)—we evaluate our interventions on OOD inputs.

Table 2: NIEs for intervention experiments, averaged over statements from sp_en_trans.

| train set | probe | LLaMA-2-13B | | LLaMA-2-70B | |
|---|---|---|---|---|---|
| | | false→true | true→false | false→true | true→false |
| cities | LR | .13 | .19 | .55 | .99 |
| | MM | .77 | .90 | .58 | .89 |
| cities+ neg_cities | LR | .33 | .52 | .61 | **1.00** |
| | MM | **.85** | **.97** | **.81** | .95 |
| | CCS | .31 | .73 | .55 | .96 |
| larger_than | LR | .28 | .27 | .61 | .96 |
| | MM | **.71** | **.79** | **.67** | 1.01 |
| larger_than+ smaller_than | LR | .07 | .13 | .54 | 1.02 |
| | MM | .26 | .53 | .66 | **1.03** |
| | CCS | .08 | .17 | .57 | 1.02 |
| likely | LR | .05 | .08 | .18 | .46 |
| | MM | .70 | .54 | .68 | .27 |

## 6.1 Experimental set-up

Let $p$ be a linear probe trained on a true/false dataset $\mathcal{D}$. Let $\boldsymbol{\theta}$ be the probe direction, normalized so that $p(\mu^- + \boldsymbol{\theta}) = p(\mu^+)$ where $\mu^+$ and $\mu^-$ are the mean representations of the true and false statements in $\mathcal{D}$, respectively; in other words, we normalize $\boldsymbol{\theta}$ so that from the perspective of the probe $p$, adding $\boldsymbol{\theta}$ turns the average false statement into the average true statement. If our model encodes the truth value of statements along the direction $\boldsymbol{\theta}$, we would expect that replacing the representation $x$ of a false statement $s$ with $x + \boldsymbol{\theta}$ would cause the model to produce outputs consistent with $s$ being a true statement.

We use inputs of the form

> The Spanish word 'fruta' means 'goat'. This statement is: FALSE
> The Spanish word 'carne' means 'meat'. This statement is: TRUE
> s. This statement is:

where s varies over sp_en_trans statements. Then for each of the probes of §5 we record:

- $PD^+$ and $PD^-$, the average probability differences $P(\text{TRUE}) - P(\text{FALSE})$ for $s$ varying over true statements or false statements in sp_en_trans, respectively,

- $PD_*^+$ and $PD_*^-$, the average probability differences where $s$ varies over true (resp. false) statements but the probe direction $\boldsymbol{\theta}$ is subtracted (resp. added) to each group (b) hidden state.

Finally, we report the *normalized indirect effects* (NIEs)

$$\frac{PD_*^- - PD^-}{PD^+ - PD^-} \quad \text{or} \quad \frac{PD_*^+ - PD^+}{PD^- - PD^+}$$

for the false→true and the true→false experiments, respectively. An NIE of 0 means that the intervention was wholly ineffective at changing model outputs; an NIE of 1 indicates that the intervention caused the LLM to label false statements as TRUE with as much confidence as genuine true statements, or vice versa.

## 6.2 Results

Results are shown in table 2. We summarize our main takeaways.

**Mass-mean probe directions are highly causal**, with MM outperforming LR and CCS in 7/8 experimental conditions, often substantially. This is true despite LR, MM, and CCS probes all have very similar sp_en_trans classification accuracies.

**Training on datasets and their opposites helps for `cities` but not for `larger_than`**. This is surprising, considering that probes trained on `larger_than` + `smaller_than` are *more* accurate on `sp_en_trans` than probes trained on `larger_than` alone (see App. D), and indicates that there is more to be understood about how training on datasets and their opposites affects truth probes.

**Training on `likely` is a surprisingly good baseline, though still weaker than interventions using truth probes.** The performance here may be due to the strong correlation ($r = .95$) between inputs being true and probable (according to LLaMA-2-70B) on `sp_en_trans`.

## 7 Discussion

### 7.1 Limitations and future work

Our work has a number of limitations. First, we focus on simple, uncontroversial statements, and therefore cannot disambiguate truth from closely related features, such as "commonly believed" or "verifiable" (Levinstein & Herrmann, 2023). Second, we study only models in the LLaMA-2 family, so it is possible that some of our results do not apply for all LLMs.

This work also raises several questions which we were unable to answer here. For instance, why were interventions with mass-mean probe directions extracted from the `likely` dataset so effective, despite these probes not themselves being accurate at classifying true/false statements? And why did mass-mean probing with the `cities` + `neg_cities` training data perform poorly poorly for the 70B model, despite mass-mean probing with `larger_than` + `smaller_than` performing well?

### 7.2 Conclusion

In this work we conduct a detailed investigation of the structure of LLM representations of truth. Drawing on simple visualizations, probing experiments, and causal evidence, we find evidence that at scale, LLMs compute and linearly represent the truth of true/false statements. We also localize truth representations to certain hidden states and introduce mass-mean probing, a simple alternative to other linear probing techniques which better identifies truth directions from true/false datasets.

## Acknowledgements

We thank Ziming Liu and Isaac Liao for useful suggestions regarding distinguishing true text from likely text, and Wes Gurnee, Eric Michaud, and Peter Park for many helpful discussions throughout this project. We thank David Bau for useful suggestions regarding the experiments in Sec. 6. Thanks also to Nora Belrose for discussion about the connection between difference-in-mean probing and linear erasure.

We also thank Oam Patel, Hadas Orgad, Sohee Yang, and Karina Nguyen for their suggestions, as well as Helena Casademunt, Max Nadeau, and Ben Edelman for giving feedback during this paper's preparation. Plots were made with Plotly (Plotly Technologies Inc., 2015). Thanks to Kevin Ro Wang for catching a typo in the statement of Thm. F.1.

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

# A   Scoping of truth

In this work, we consider declarative factual statements, for example "Eighty-one is larger than fifty-four" or "The city of Denver is in Vietnam." We scope "truth" to mean factuality, i.e. the truth or falsehood of these statements; for instance the examples given have truth values of true and false, respectively. To be clear, we list here some notions of "truth" which we do not consider in this work:

- Correct question answering (considered in Li et al. (2023b) and for some of the prompts used in Burns et al. (2023)). For example, we do not consider "What country is Paris in? France" to have a truth value.

- Presence of deception, for example dishonest expressions of opinion ("I like that plan").

- Compliance. For example, "Answer this question incorrectly: what country is Paris in? Paris is in Egypt" is an example of compliance, even though the statement at the end of the text is false.

Moreover, the statements under consideration in this work are all simple, unambiguous, and uncontroversial. Thus, we make no attempt to disambiguate "true statements" from closely-related notions like:

- Uncontroversial statements

- Statements which are widely believed

- Statements which educated people believe.

On the other hand, our statements *do* disambiguate the notions of "true statements" and "statements which are likely to appear in training data"; See our discussion at the end of §2.

# B   Full patching results

Fig. 6 shows full patching results. We see that both LLaMA-2-7B and LLaMA-2-13B display the "summarization" behavior in which information relevant to the full statement is represented over the end-of-sentence punctuation token. On the other hand, LLaMA-2-70B displays this behavior in a context-dependent way – we see it for `cities` but not for `sp_en_trans`.

# C   Emergence of linear structure across layers

The linear structure observed in §4 follows the following pattern: in early layers, representations are uninformative; then, in early middle layers, salient linear structure in the top few PCs rapidly emerges, with this structure emerging later for statements with a more complicated logical structure (e.g. conjunctions). This is shown for LLaMA-2-13B in Fig. 7. We hypothesize that this is due to LLMs hierarchically developing understanding of their input data, progressing from surface level features to more abstract concepts.

The misalignment in Fig. 3(c) also has an interesting dependence on layer. In Fig. 8 we visualize LLaMA-2-13B representations of `cities` and `neg_cities` at various layers. In early layers (left) we see *antipodal* alignment as in Fig. 3(b, center). As we progress through layers, we see the axes of separation rotate to lie orthogonally, until they eventually align.

One interpretation of this is that in early layers, the model computed and linearly represented some feature (like "close association") which correlates with truth on both `cities` and `neg_cities` but with opposite signs. In later layers, the model computed and promoted to greater salience a more abstract concept which correlates with truth across both datasets.

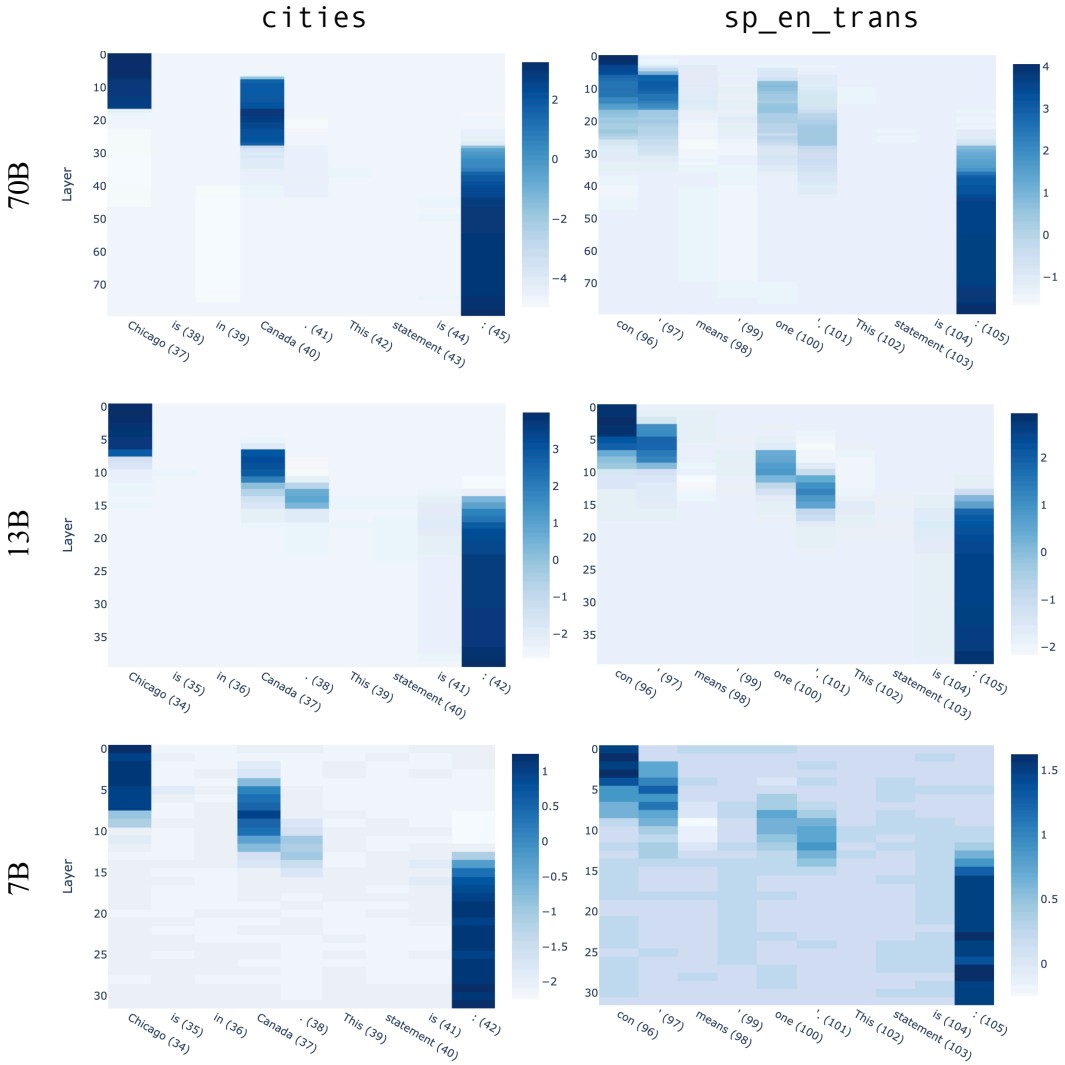

Figure 6: Full patching results across all three model sizes and inputs. Results are for patching false inputs (shown) to true by changing the first token shown on the left. Numbers in parentheses are the index of the token in the full (few-shot) prompt.

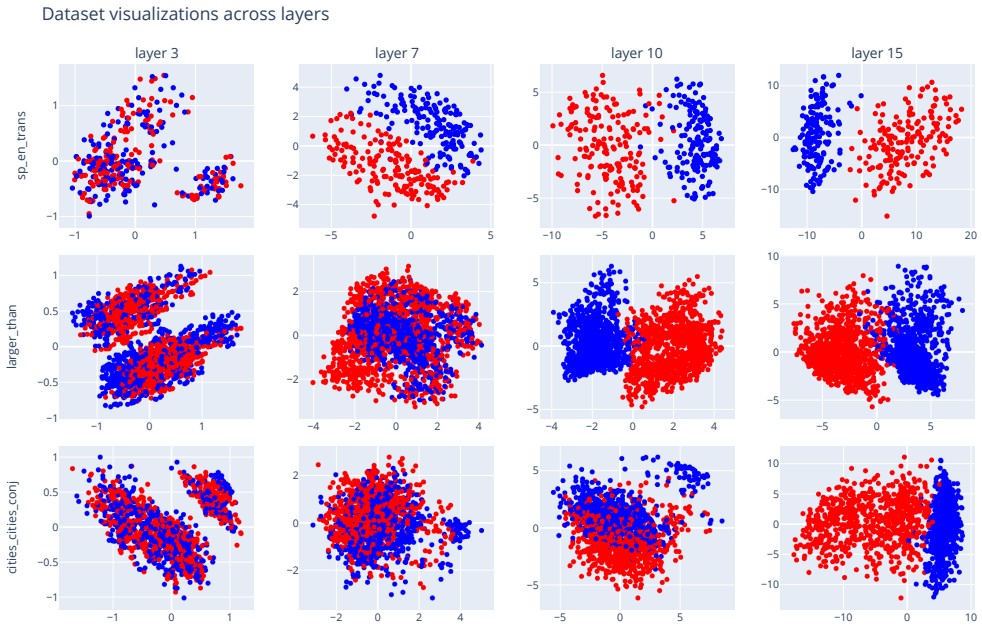

Figure 7: Projections of LLaMA-2-13B representations of datasets onto their top two PCs, across various layers.

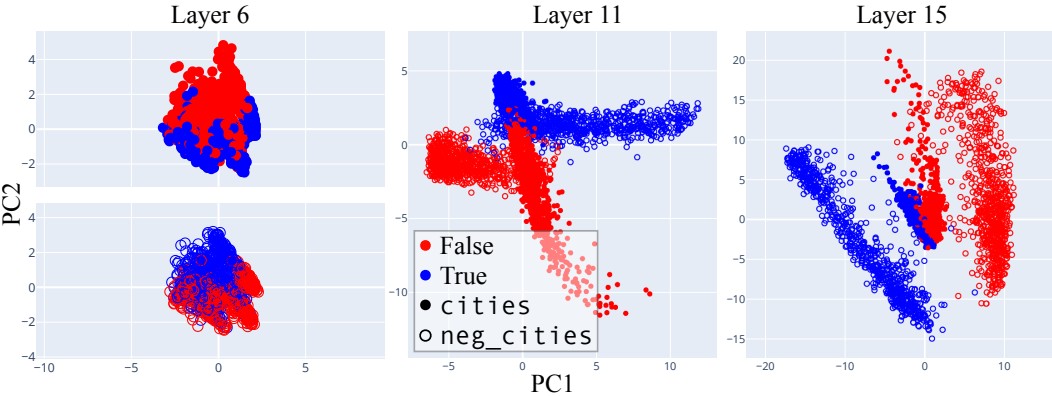

Figure 8: PCA visualizations of LLaMA-2-13B representations of cities and neg_cities at various layers.

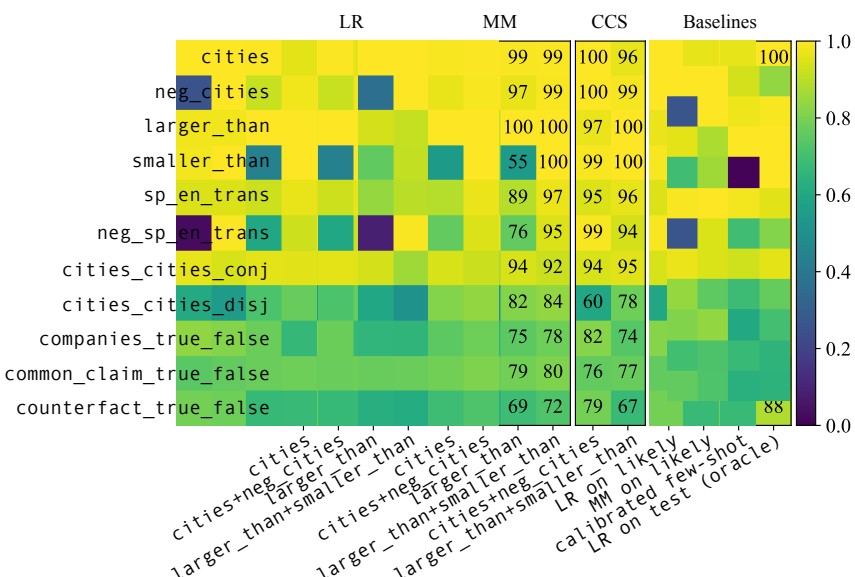

Figure 9: Generalization results for **LLaMA-2-70B**.

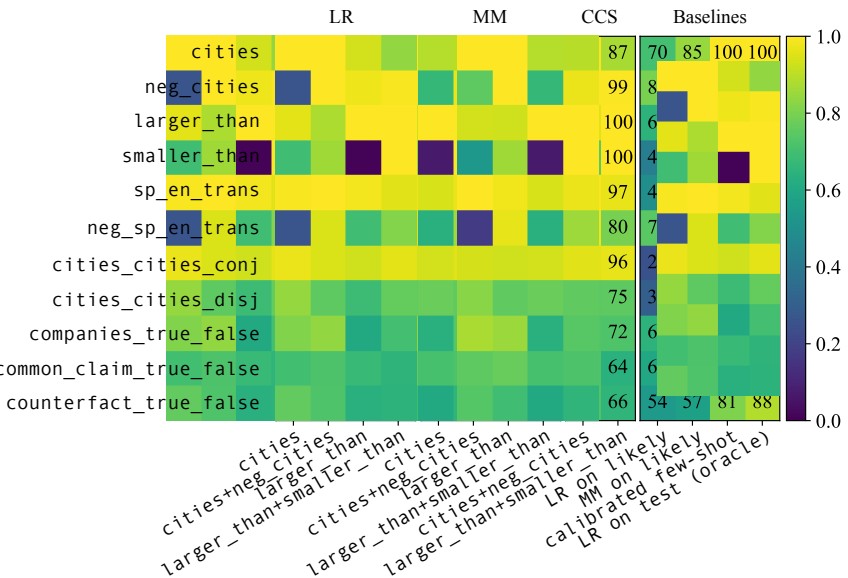

Figure 10: Generalization results for **LLaMA-2-13B**.

## D   Full generalization results

Here we present the full generalization results for probes trained on LLaMA-2-70B (Fig. 9), 13B (Fig. 10), and 7B (Fig. 11). The horizontal axis shows the training data for the probe and the vertical axis shows the test set.

## E   Mass-mean probing in terms of Mahalanobis whitening

One way to interpret the formula $p_{\text{mm}}^{\text{iid}}(x) = \sigma(\theta_{\text{mm}}^T \Sigma^{-1} x)$ for the IID version of mass-mean probing is in terms of Mahalanobis whitening. Recall that if $\mathcal{D} = \{x_i\}$ is a dataset of $x_i \in \mathbb{R}^d$ with covariance matrix $\Sigma$, then the Mahalanobis whitening transformation $W = \Sigma^{-1/2}$

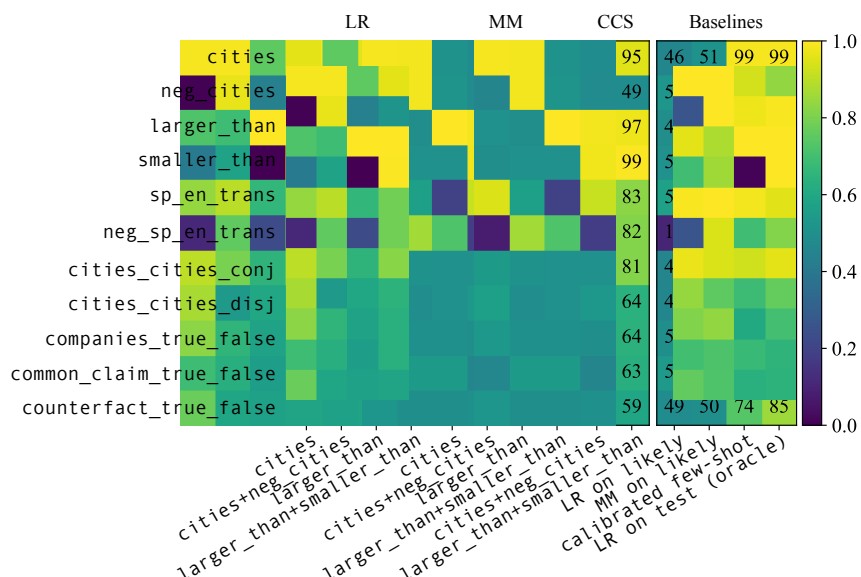

Figure 11: Generalization results for **LLaMA-2-7B**.

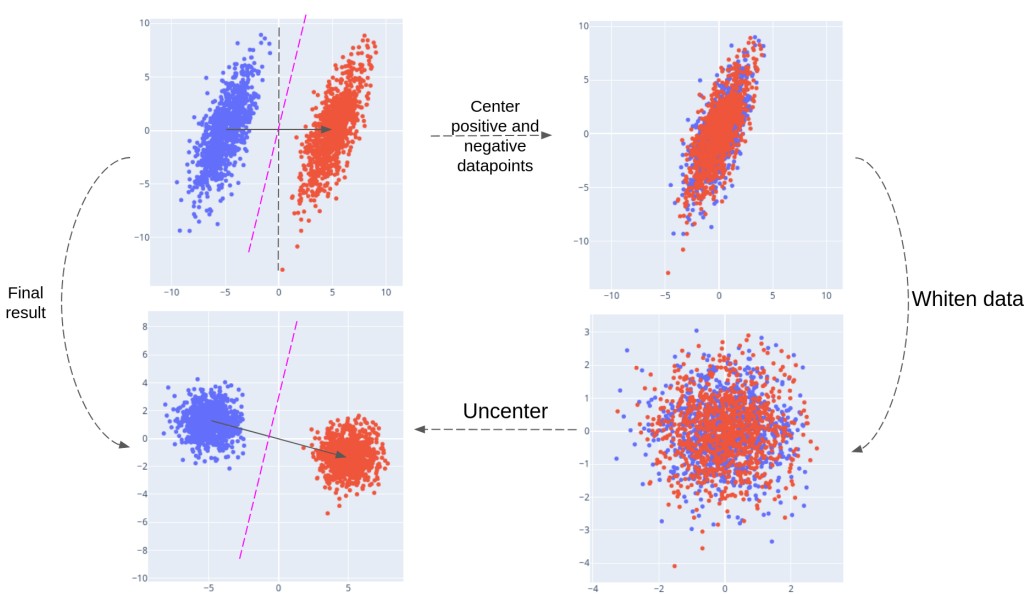

Figure 12: Mass-mean probing is equivalent to taking the projection onto $\theta_{\mathrm{mm}}$ after applying a whitening transformation.

satisfies the property that $\mathcal{D}' = \{Wx_i\}$ has covariance matrix given by the identity matrix, i.e. the whitened coordinates are uncorrelated with variance 1. Thus, noting that $\theta_{\mathrm{mm}}^T \Sigma^{-1} x$ coincides with the inner product between $Wx$ and $W\theta$, we see that $p_{\mathrm{mm}}$ amounts to taking the projection onto $\theta_{\mathrm{mm}}$ after performing the change-of-basis given by $W$. This is illustrated with hypothetical data in Fig. 12.

## F  For Gaussian data, IID mass-mean probing coincides with logistic regression on average

Let $\theta \in \mathbb{R}^d$ and $\Sigma$ be a symmetric, positive-definite $d \times d$ matrix. Suppose given access to a distribution $\mathcal{D}$ of datapoints $x \in \mathbb{R}^d$ with binary labels $y \in \{0,1\}$ such that the negative datapoints are distributed as $\mathcal{N}(-\theta, \Sigma)$ and the positive datapoints are distributed as $\mathcal{N}(\theta, \Sigma)$. Then the vector identified by mass-mean probing is $\theta_{\mathrm{mm}} = 2\theta$. The following theorem then shows that $p_{\mathrm{mm}}^{\mathrm{iid}}(x) = \sigma(2\theta^T \Sigma^{-1} x)$ is also the solution to logistic regression up to scaling.

**Theorem F.1.** *Let*

$$\theta_{\mathrm{lr}} = \underset{\phi: \|\phi\|=1}{\arg \min} \, \mathbb{E}_{(x,y) \sim \mathcal{D}} \left[ y \log \sigma \left( \phi^T x \right) + (1-y) \log \left( 1 - \sigma \left( \phi^T x \right) \right) \right]$$

*be the direction identified by logistic regression. Then $\theta_{\mathrm{lr}} \propto \Sigma^{-1} \theta$.*

*Proof.* Since the change of coordinates $x \mapsto Wx$ where $W = \Sigma^{-1/2}$ (see App. E) sends $\mathcal{N}(\pm\theta, \Sigma)$ to $\mathcal{N}(\pm W\theta, I_d)$, we see that

$$W\Sigma\theta_{\mathrm{lr}} = \underset{\phi: \|\phi\|=1}{\arg \min} \, \mathbb{E}_{(x,y) \sim \mathcal{D}'} \left[ y \log \sigma \left( \phi^T Wx \right) + (1-y) \log \left( 1 - \sigma \left( \phi^T Wx \right) \right) \right]$$

where $\mathcal{D}'$ is the distribution of labeled $x \in \mathbb{R}^d$ such that the positive/negative datapoints are distributed as $\mathcal{N}(\pm W\theta, I_d)$. But the argmax on the right-hand side is clearly $\propto W\theta$, so that $\theta_{\mathrm{lr}} \propto \Sigma^{-1}\theta$ as desired. $\square$

## G  Difference-in-means directions and linear concept erasure

In this appendix, we explain the connection between difference-in-means directions and optimal erasure. One consequence of this connection is that it suggests a natural extension of difference-in-means probes to multi-class classification data.

The connection comes via the following theorem from Belrose et al. (2023).

**Theorem G.1.** *(Belrose et al., 2023, Thm. 3.1.) Let $(X, Y)$ be jointly distributed random vectors with $X \in \mathbb{R}^d$ having finite mean and $Y \in \mathcal{Y} = \{\mathbf{y} \in \{0,1\}^k : \|\mathbf{y}\|_1 = 1\}$ (representing one-hot encodings of a multi-class labels). Suppose that $\mathcal{L} : \mathbb{R}^k \times \mathcal{Y} \to \mathbb{R}^{>0}$ is a loss function convex in its first argument (e.g. cross-entropy loss).*

*If the class-conditional means $\mathbb{E}[X|Y=i]$ for $i \in \{1, \dots, k\}$ are all equal, then the best affine predictor (that is, a predictor $\eta : \mathbb{R}^d \to \mathbb{R}^k$ of the form $\eta(x) = Wx + b$) is constant $\eta(x) = b$.*

In the case of a binary classification problem $(X, Y)$, this theorem implies that any nullity 1 projection $P$ which eliminates linearly-recoverable information from $X$ has kernel

$$\ker P = \mathrm{span}(\delta)$$

generated by the difference-in-mean vector $\delta = \mu^+ - \mu^-$ for the classes.

For a more general multi-class classification problem, one could similarly ask: What is the "best" direction to project away in order to eliminate linearly-recoverable information from $X$? A natural choice is thus the top left singular vector of the cross-covariance matrix $\Sigma_{XY}$. (In the case of binary classification, we have that $\Sigma_{XY} = [-\delta \; \delta]$ has column rank 1, making $\delta$ the top left singular vector.)

# H  Details on dataset creation

Here we give example statements from our datasets, templates used for making the datasets, and other details regarding dataset creation.

**cities.** We formed these statements from the template "The city of [city] is in [country]" using a list of world cities from Geonames (2023). We filtered for cities with populations $> 500,000$, which did not share their name with any other listed city, which were located in a curated list of widely-recognized countries, and which were not city-states. For each city, we generated one true statement and one false statement, where the false statement was generated by sampling a false country with probability equal to the country's frequency among the true datapoints (this was to ensure that e.g. statements ending with "China" were not disproportionately true). Example statements:

- The city of Sevastopol is in Ukraine. (TRUE)
- The city of Baghdad is in China. (FALSE)

**sp_en_trans.** Beginning with a list of common Spanish words and their English translations, we formed statements from the template "The Spanish word '[Spanish word]' means '[English word]'." Half of Spanish words were given their correct labels and half were given random incorrect labels from English words in the dataset. The first author, a Spanish speaker, then went through the dataset by hand and deleted examples with Spanish words that have multiple viable translations or were otherwise ambiguous. Example statements:

- The Spanish word 'imaginar' means 'to imagine'. (TRUE)
- The Spanish word 'silla' means 'neighbor'. (FALSE)

**larger_than and smaller_than.** We generate these statements from the templates "x is larger than y" and "x is smaller than y" for $x, y \in \{$fifty-one, fifty-two,...,ninety-nine$\}$. We exclude cases where $x = y$ or where one of x or y is divisible by 10. We chose to limit the range of possible values in this way for the sake of visualization: we found that LLaMA-13B linearly represents the size of numbers, but not at a consistent scale: the internally represented difference between one and ten is considerably larger than between fifty and sixty. Thus, when visualizing statements with numbers ranging to one, the top principal components are dominated by features representing the sizes of numbers.

**neg_cities and neg_sp_en_trans.** We form these datasets by negating statements from cities and sp_en_trans according to the templates "The city of [city] is not in [country]" and "'The Spanish word '[Spanish word]' does not mean '[English word]'."

**cities_cities_conj and cities_cities_disj.** These datasets are generated from cities according to the following templates:

- It is the case both that [statement 1] and that [statement 2].
- It is the case either that [statement 1] or that [statement 2].

We sample the two statements independently to be true with probability $\frac{1}{\sqrt{2}}$ for cities_cities_conj and with probability $1 - \frac{1}{\sqrt{2}}$ for cities_cities_disj. These probabilities are selected to ensure that the overall dataset is balanced between true and false statements, but that there is no correlation between the truth of the first and second statement in the conjunction.

**likely.** We generate this dataset by having LLaMA-13B produce unconditioned generations of length up to 100 tokens, using temperature 0.9. At the final token of the generation, we either sample the most likely token or the 100th most likely final token. We remove generations which contain special tokens. Dataset examples:

- The 2019-2024 Outlook for Women's and Girls' Cut and Sew and Knit and Crochet Sweaters in the United States This study covers the latent demand outlook for (LIKELY)

- Tags: python, django Question: How to get my django app to work with python 3.7 I am new to django and have been trying to install it in my pc. I have installed python 3.7 together (UNLIKELY)

**companies_true_false.** This dataset was introduced by Azaria & Mitchell (2023); we obtained it via the project repository for Levinstein & Herrmann (2023) which also used the dataset. Example statements:

- ArcelorMittal has headquarters in Luxembourg. (TRUE)
- Exxon Mobil engages in the provision of banking and financial services. (FALSE)

**common_claim_true_false.** CommonClaim was introduced in Casper et al. (2023). It consists of various statements generated by GPT-3-davinci-002, labeled by humans as being true, false, or neither. If human labelers disagreed on the truth of a statement, this is also recorded. We adapted CommonClaim by selecting statements which were labeled true or false with no labeler disagreement, then removing excess true statement to balance the dataset. Example statements:

- Tomatoes are not actually a vegetable. (TRUE)
- Contrary to popular belief, the platypuses are not venomous. (FALSE)

As these examples show, the statements can be ambiguous or of unclear truth value.

**counterfact_true_false.** Counterfact was introduced in Meng et al. (2022) and consists of factual recall statements. We adapt Counterfact by using statements which form complete sentences and, for each such statement, using both the true version and a false version given by one of Counterfact's suggested false modifications. We also append a period to the end. Example statements:

- Olaus Rudbeck spoke the language Swedish. (TRUE)
- The official religion of Malacca sultanate is Christianity. (FALSE)

