# OpenReview forum: "The Geometry of Truth: Emergent Linear Structure in Large Language Model Representations of True/False Datasets"
_colmweb.org/COLM/2024/Conference — COLM_

### Official Review · Reviewer_Pvmb · 2024-04-24

**Rating:** 7
**Confidence:** 4
**Ethics Flag:** 1

**Summary:**

This work presents a series of three experiments which are designed to better understand how the truthfulness of factual statements is encoded in LLMs. The authors conclude that whether a factual statement is true or false is encoded linearly in the intermediate hidden states, and that this linear separability is emergent with scale (i.e., is present in relatively large models, but not smaller models). This conclusion is supported by PCA visualizations of hidden states (Sec. 4), probing of hidden states (Sec. 5), and causal interventions of hidden states (Sec. 6).

The experiments specifically focus on the LLama 2 models and use novel datasets of facts created with simple templates for evaluation. The statements are for instance “The city of Chicago is in Canada.” where the particular hidden state being studied is some middle layer representation of the final punctuation token. (This choice of hidden state is motivated by a preliminary “patching” experiment.) Linear probes are trained on the hidden state and the generalization of these probes is tested across out-of-domain factual statements. Finally, hidden states are modified by adding a normalized vector in the direction of truthfulness found in the linear probes, and the resulting prob. of predicting “This statement is: [TRUE/FALSE]” is studied.

Overall, I found the experiments straightforward and the main conclusion that linear separability increases with model size interesting. Linear probing for truthfulness of factual statements is not novel, however; Li et al. (2023b) and Burns et al. (2023) have already provided evidence that truthfulness may be encoded linearly in LM hidden states. The paper proposes some potential weaknesses in terms of the generalizability of these earlier conclusions, but the external validity of the paper is itself limited by the scope of the datasets used. Therefore, the main contribution seems to be a somewhat incremental extension of earlier work by studying linear separability at multiple model sizes. It is also not clear to me what can be learned from the emphasis on Linear Discriminant Analysis (LDA) as a new probing method (called mass-mean probing in the paper).

Edit: Based on the author response and the points raised in the other reviews, I do believe the contribution is significant enough to be of interest to the community. Furthermore, Li et al. (2023b) was published at NeurIPS 2023 and may therefore be considered contemporaneous work.

**Questions To Authors:**

Questions:
1. If mass-mean pooling is equivalent to Fisher’s Linear Discriminant Analysis, why not use this existing terminology? E.g., why not use the more common abbreviations FLDA or LDA rather than MM?
1. What is the intended meaning of the following statements regarding LLMs in the introduction? I’m not sure I understand the connection to the paper: i) “In some cases, this is because they do not know better.” ii) “In other cases, LLMs apparently know that statements are false
but generate them anyway.”

Minor typos:
* Missing word: “Second, specifically interrogate whether our probes attend to truth, rather than merely features which correlate with truth (e.g. probable vs. improbable text).”
* Citation formatting: “the maximum margin separator Soudry et al. (2018)”
* Capitalization: (FISHER, 1936)
* Extra comma: “The superposition hypothesis of Elhage et al. (2022), suggests this may be typical in deep networks.”
* Duplicated word: “And why did mass-mean probing with the cities + neg cities training data perform poorly poorly for the 70B model”

**Reasons To Accept:**

* The paper presents the novel conclusion that the linear separability of the truthfulness of factual statements increases with LM size.
* The paper presents ways to improve the robustness of earlier conclusions regarding the linear separability of truthfulness by focusing on targeted statements, out-of-domain generalization, and the probing mechanism.
* The experiments are clearly described.

**Reasons To Reject:**

* The contribution seems incremental. My understanding of the main contributions being presented are: 1) the new dataset of factual statements; 2) new analysis of how linear separability is related to scale; 3) a novel probing method based on Linear Discriminant Analysis (LDA) called mass-mean pooling. However, the datasets are simple templated examples that I do not consider a significant contribution, and the LDA probing method performs very similarly to logistic regression in terms of generalization out-of-domain (Figure 5b). Therefore, the main contribution seems to be expanding existing analysis of truthfulness to multiple model scales.
* While the work is presented as studying truthfulness, the limited scope of the main “curated” datasets means that what is being measured is underspecified and may not be truthfulness at all. (E.g., it could be that “commonly believed” or “verifiable” declarative statements are linearly separable as mentioned in the limitations section.) And yet, a motivation given in the paper is that prior work which already showed that "truthfulness" is linearly separable was limited in scope. I believe this weakness could be improved by more explicitly stating exactly where previous work is limited in the scope of conclusions, and exactly what results address in the paper address said limitations.

---

> ### Author Rebuttal · Authors · 2024-05-31
>
> Thank you for your thoughtful review.
>
> The reviewer’s main concern is whether our work is a significant contribution beyond prior work. This is discussed briefly in paragraph 2 of Sec. 1.1, but we’ll go into more detail here:
> 1. **A detailed study of generalization.** Our main finding is *not* that linear separation of true/false statements emerges with scale. Rather, it’s that at scale, there is better *generalization* of truth directions. This is key for our argument: if datasets were individually linearly separable but not along a common direction, then we could not conclude that LLMs have a general truth representation. It is also crucially missing from prior work: while [1,2] study generalization, [1] finds negative results and [2]—which works with models at the GPT-2 scale or smaller—finds negative results for autoregressive LMs (see Fig. 9). Our demonstration that truth directions generalize well at scale is novel.
> 2. **Probing causality.** As the reviewer points out, mass-mean probing has similar classification performance to logistic regression (LR). However, we also compare probing techniques on how causally implicated they are and find that LR directions are not as causal; this is evidence that LR probes are worse for identifying truth directions. There are hints of this in [2], but they do not study their probes OOD, and their setting is not clearly about truth. (Their data consists of intentionally confusing questions for which around 15% have “No comment” as the “truthful” answer!)
>
> Please let us know if you have more questions about how our work improves on prior work. If accepted, we’ll use the additional camera-ready page to discuss this in much more detail.
>
> Re terminology: note that only the in-distribution variant of mass-mean probing is equivalent to LDA. When evaluating OOD, it is a difference-of-means probe with no covariance correction. The point is that the argument for using mass-mean probing in Sec. 5.1 only suggests that the covariance correction will help in-distribution (and might make things worse OOD). We will clarify this.
>
> Re Q2: Some researchers believe that an LLM would only generate a falsehood F if the LLM believes F is true (or otherwise doesn’t understand the difference between true and false). Here we are trying to clarify that LLMs can understand that F is false, but still generate F (as illustrated with the examples in the introduction).
> [1] https://arxiv.org/abs/2307.00175
> [2] https://arxiv.org/abs/2212.03827

---

> > ### Comment · Reviewer_Pvmb · 2024-06-03
> > **Review response**
> >
> > Thank you for your response. I have updated my review accordingly.
> >
> > I suggest to more explicitly distinguish the presented work from earlier work in the introduction of the paper, incorporating those points raised in the rebuttal. I also believe the paper would benefit from emphasizing the mentioned point of how mass-mean probing differs from LDA.
> >
> > I still strongly agree with reviewer jLEs that the curated dataset is not a significant contribution of the work, and I believe it would make sense to put less emphasis on the dataset as a contribution.
> >
> > Finally, I'm still not sure I exactly understand your answer to Q2. My suggestion would be to use more explicit wording e.g. possibly change "the LLM believes F is true" to "the truthfulness of F is linearly encoded in the hidden state of the LLM" to make it more clear what "believes" is being used to refer to here. The current wording is likely ambiguous.

---

### Official Review · Reviewer_UMfD · 2024-04-28

**Rating:** 7
**Confidence:** 4
**Ethics Flag:** 1

**Summary:**

The paper curates a data set for probing TRUE and FALSE believes of LLMs and uses this datasets to provide evidence that LLMs of the LLama2 family reveal a clear linear structure separating TRUE from FALSE statements that can be transferred across data sets and can be causally intervened. The study further tries to correct of general probability and likelihood or other features that may intervene.

**Reasons To Accept:**

- Well structured and written work that tries to find evidence to what extent believes are actually modeled and therefore models have awareness of truth/falsehood
- The different data sets allow for a precise investigation of possible factors without too many intervening factors. The paper can be inspiring for the development of many more of such datasets.

**Reasons To Reject:**

- The simplicity and clarity of the datasets also has a disadvantage that it does not reflect models behaviour in complex contexts and for more nuanced epistemic values than just TRUE and FALSE, such (im)possible, rare, likely)
- What does it really mean that a model “believes” a statement to be true or false? This also relates to the previous point that being false and impossible is different from being false and possible. There should be more discussion on this and also what this implies for using LLMs.

---

> ### Author Rebuttal · Authors · 2024-05-31
>
> Thank you for your review. We’re glad you found the paper well structured.
>
> > The simplicity and clarity of the datasets also has a disadvantage that it does not reflect models behaviour in complex contexts and for more nuanced epistemic values than just TRUE and FALSE
>
> Yes, this is certainly a limitation of the paper. One motivation for this work is that there was —and still is—controversy around whether LLMs understand the notion of true vs. false at all, with many researchers asserting that LLMs only seem to understand this in cases where true statements are more probable than false ones (in the sense of being more likely to appear in the LM’s training distribution). We felt that the most important next step was to establish that models internally represent the difference between true and false statements in a way that goes beyond just probable vs. improbable. To make this point as compellingly as possible, we restricted to data which captures a narrow, unambiguous notion of true vs. false. We’re excited for future work to study these more nuanced gradations of “truth”!
>
> > What does it really mean that a model “believes” a statement to be true or false? This also relates to the previous point that being false and impossible is different from being false and possible. There should be more discussion on this and also what this implies for using LLMs.
>
> In the paper we almost entirely avoid using the word “believe,” except for when motivating the problem in the introduction. But since you ask about it, we view our work as a first step towards giving an empirical grounding for a way to define “believes”: that there is some internal difference between the way certain classes of statements are represented which is causally implicated in whether the model produces natural-language outputs consistent with treating those statements as true. For example if, as our paper suggests, there is a linear representation of truth, then you should be able to take LM’s the representation of “The city of Chicago is in China,” shift it along the truth direction, and thereby induce the model to produce outputs consist with treating “The city of Chicago is in China” as a true statement (as we show you can do).
>
> This is important because if there is a general representation of truth, then it should be possible to build LM “lie detectors” which can generalize to novel settings, with true/false statements unlike those we used to train our lie detector.

---

> > ### Comment · Reviewer_UMfD · 2024-06-04
> >
> > It would be nice to put the work in a wider perspective in the discussion or conclusion that this could be the basis for epistemic models in the future. Except for that I am happy with the discussion and response.

---

### Official Review · Reviewer_jLEs · 2024-05-09

**Rating:** 6
**Confidence:** 4
**Ethics Flag:** 1

**Summary:**

This paper explores the representation of truth in the latent representation of Large Language Models (LLMs) using true/false datasets. The work examines the linear structure in LLM representations through visualizations, transfer experiments, and causal interventions. The study introduces a new probing technique called mass-mean probing, which identifies causally implicated directions in model outputs. The findings suggest that LLMs can linearly represent the truth of factual statements at scale, and the paper localizes truth representations to specific hidden states.
The authors conduct a novel study on a specific use case - the linearity of the concept of truthfulness - however, the paper fails in our opinion to clearly explain the actual implication of discovering such evidence. The significance, and contributions of this work, is therefore considered rather weak by this reviewer.

**Questions To Authors:**

1. Can you provide more information on the dataset, its creation and the motivation of not using well-known datasets such as FEVER?
2. Can you provide more information about the training process of the probes?

**Reasons To Accept:**

1. The work represents a novel and interesting study of the linearity of the concept of truthfulness in the latent representation of Large Language Models.
2. The authors introduce a new probing technique called mean probing to better identify the directions which are most causally implicated.

**Reasons To Reject:**

1. The authors claim, as one of their contributions, the curation of a high-quality datasets of true / false factual statements which are uncontroversial, unambiguous, and simple. They however do not clearly explain how they create it, how they assess factuality, and most importantly, how they validate the claimed properties (e.g., unambiguous). In addition, this reviewer is not convinced about the value of its actual contribution, as there exist several well-known fact verification datasets in the literature -- such as FEVER (Thorne et al., 2018), VitaminC (Schuster et al., 2021) and CREAK (Onoe et al., 2021).
2. In section 3, the authors measure the difference in log probability between the tokens “True” and “False"; however, this reviewer does not see clear evidence for their empirical hypothesis: from a weak interpretation of what the model encodes (e.g., the group “b” encodes information pertaining to the full statement) to the choice of focusing on middle hidden layers (i.e., group b).
3. Although the authors elaborate on the differences between theirs and previous related works, the paper does not clearly explain the motivations and implications of demonstrating linearity and its direction. In section 4, the authors show linear structures in the hidden state (chosen arbitrarily?) with a simple dimensionality reduction technique (i.e., PCA). It does not seem clear how this generalizes, and what the implications are. A deeper analysis of evolution across layers (Appendix C) would have been, in our opinion, more valuable.
4. The organization of the paper is not satisfactory, this reviewer suggests a clearer separation between methods, experiments, results and discussions to improve readability.

---

> ### Author Rebuttal · Authors · 2024-05-31
>
> Thank you for your thoughtful review.
> > W1/Q1: Value of the dataset
>
> Re dataset construction: App. H details our process. E.g. for the cities dataset we used a world cities database, removed cities that share a name with another city/country, filtered for cities of population >500,000, filtered for universally-recognized countries, and finally used the remaining data to produce statements of the form “The city of [city] is in [country].” Our datasets can be seen at https://anonymous.4open.science/r/geometry-of-truth-8237/datasets/cities.csv
>
> Re prior datasets: we conducted a literature review for datasets that approximately met our desiderata, and adapted those we found into our uncurated datasets (Sec. 2). Many other datasets capture factuality-adjacent phenomena: FEVER and VitaminC for checking whether claims are textually supported, CREAK for commonsense reasoning, and many datasets for question-answering. These datasets were unsuitable for our use case:
> They are challenging, but we want easy data: instead of benchmarking model accuracy for classifying true/false statements, we want to use data that models can reliably classify as true/false to study LM knowledge representations.
> They capture phenomena distinct from factuality. We did not want to assume e.g. that models represent true vs. false and textually supported vs. unsupported in similar ways.
>
> If accepted, we will use the additional camera-ready page to discuss our datasets in much more detail.
> > W2: Choice of hidden states
>
> Sec. 3 is not itself strong evidence that the group (b) hidden states encode a representation of the truth—see Secs. 4-6 for that. Rather, Sec. 3 rules out all *other* hidden states, either because they are not causal, or because they can be shown to encode other information. We will clarify this.
> > W3: Motivation of our work
>
> Our line of work studies the extent to which LMs have world models, a question of substantial scientific interest in the ML community [1], with special controversy around abstractions like “truth” in particular [2]. We felt that finding a direction which is causal for factuality across diverse settings would be a valuable contribution to this discussion.
> > Q2: Details about probes
>
> What additional information about the probes’ training beyond that in Secs. 5.1-5.2 is the reviewer interested in? We would be happy to give additional clarifications here or in a revision.
> [1] https://openreview.net/forum?id=DeG07_TcZvT
> [2] https://arxiv.org/abs/2307.00175

---

> > ### Comment · Reviewer_jLEs · 2024-06-05
> >
> > Thank you for your clarifications.
> > I have updated my evaluation on the basis of your rebuttal.
> > I have a few additional comments regarding the dataset construction: I suggest presenting it as a collection of prompts for with true/false claims, which aligns with your primary goal, rather than listing it as one of the paper's contributions. Its creation is described in an appendix towards the end of the manuscript, and both the creation process and the knowledge base used seem relatively basic, not representing in my opinion a core contribution of the paper.

---

### Official Review · Reviewer_VXXD · 2024-05-11

**Rating:** 8
**Confidence:** 3
**Ethics Flag:** 1

**Summary:**

This paper investigates whether the concept of truth (operationalized as unambiguously factual vs. nonfactual statements in the real world) is linearly represented in the Llama-2 family of models via transferability of of behavioral accuracy and representational intervention success across sets of examples that vary in structure and topic.

- Quality: The claims being made are generally well argued on the basis of convincing experimental results. I found the results showing generalization from training on larger than + smaller than to both negated and un-negated versions of the test datasets to be convincing in particular.
- Clarity: Presentation and writing are generally clear. The dataset domains and design decisions could be more clearly motivated.
- Originality: Makes original contributions in terms of the empirical results presented, dataset, and probing method. Linear separability of "truthfulness" itself is not topically novel, but the work claims to address a gap that has not been systematically explored in prior work (uncontroversial factuality)
- Significance: The paper makes multiple contributions that the COLM community is likely to be interested in, and leaves open several directions that can be followed up by future work. One of the key limitations, as the authors also explicitly acknowledge, is the narrow range of models evaluated which may limit the generalizability of the findings.

**Questions To Authors:**

- Clarification about setting: "Unlike in §3, we do not prepend the statements with a few-shot prompt (so our models are not “primed” to consider the truth value of our statements).". Is it still post-pended with "This statement is:"? I wonder if this still provides some sort of structural scaffolding and whether this linear representation of truth emerges without any sort of "structural packaging". For instance, we may imagine a scenario where the _linear_ representation of truth only emerges only under structural packaging of the target statements, which is slightly more nuanced than saying "LLMs linearly represent the truth or falsehood of factual statements" (from the abstract) in general.
- I appreciated the effort to disentangle probable vs. truthful. This is currently done by using negation, but I think people will also be interested in the TruthfulQA type of cases where a misalignment between probable-ness and truthfulness arise due to common misconceptions - it would be interesting to see if the current findings generalize to such cases, although the authors are currently framing the focus on uncontroversial examples as a contribution (and I agree that it is). I would assume generalization experiments that are similar to the ones presented in this work should be quite straightforward to run on these kinds of examples (perhaps constructed or reworded from existing datasets to match stylistic features.
- What is the source of the consistent gaps in performance across directions of intervention (true -> false vs. false vs. true)? The NIE being larger for true -> false than false -> seems to be consistently observed across scale.
- "concept of truth emerging" in larger models vs. the models having the factual knowledge to even recognize that something is true vs. false seem like two different questions. The current interpretation of the different results across scale is that larger models represent truth linearly, but smaller models don't. But it could be that smaller models also do represent truth linearly but just falls short on factual knowledge on more examples within the dataset. This consideration seems already somewhat baked into the design principle for the datasets tested ("Our curated datasets consist of statements that are uncontroversial, unambiguous, and simple enough that our LLMs are likely to understand them."), but I'm wondering if it would make sense to actually provide some basis for this claim, for instance by simply checking the model accuracy on the statements via the most probable completion ("The city of Zagreb is in " - the LM can be considered to have this knowledge if Croatia is the most probable completion by the LM). But constructing such tests for negated statements would not be as straightforward.
- Added after reading other reviewers' comments: I agree that the domains of the datasets and the design decisions could be motivated better. In particular, I found the decision to select a multilingual domain somewhat odd (given that Llama 2 isn't explictly multilingual, at least not in a way that's properly benchmarked) and could lead to possible confounds in ways relevant to the comment above.

**Reasons To Accept:**

This is an overall solid paper that makes contributions along multiple dimensions (empirical findings, evaluation dataset, and probing method). I found the arguments and supporting evidence to be generally convincing.Added after reading other reviewers' comments: I do agree that the motivation behind the dataset creation and design could use more discussion.

**Reasons To Reject:**

No strong reasons to reject except the narrow range of models tested. I could see there being a lot of disagreements about the operationalization for "truth" and whether the proposed dataset captures it well/what kind of definition we _should_ be adopting, and so forth. Nevertheless, I think the work has value in that they explore a setting that has not been explored in prior work and there are interesting findings within that scope. The motivation for the particular domains of datasets selected could also be more clearly discussed.

---

> ### Author Rebuttal · Authors · 2024-05-31
>
> Thank you for your review—we’re glad that you think our paper makes multiple contributions of interest to the COLM community.
>
> Q1: In sections 3-4, the model is *only* presented with a single, raw statement. “This statement is” is *not* postpended, so the model is not primed at all to consider the factuality of the statement.
>
> Q2: We agree that this would be interesting. We’ll note that many of the statements in the CommonClaim dataset we evaluate generalizations to have a similar character to TruthfulQA questions, e.g.:
> * Bears are the only mammals in North America known to prey on humans. (False)
> * Opera was once magical entertainment for the elegant elite. (True)
>
> As shown in Fig. 9, for LLaMA-2-70B probes trained on a curated dataset are generally 75-80% accurate on CommonClaim.
>
> Q3: This is a great question! Sadly, we’re not sure why this is.
>
> Q4:
> > it could be that smaller models also do represent truth linearly but just falls short on factual knowledge on more examples within the dataset
>
> We do not think this is consistent with our results. If this were the case, then for small models we would see low performance both in- and out-of-distribution. But instead we see high in-distribution performance and low out-of-distribution performance. This indicates that small models know enough to distinguish the true from false statements for each data distribution individually, but that they do not cohere this information into a unified concept of “true vs. false.” If you agree, we’ll try to clarify this in the text.
>
> > simply checking the model accuracy on the statements via the most probable completion
>
> Please note that we do this—see our “calibrated few-shot prompting” baseline.
>
>
> Q5: Re Spanish-English translation, Spanish Wikipedia is part of the training data for LLaMA models [1]. Generally, our goal was to statements with good topical *and* structural diversity. E.g. the cities dataset is a factual recall task, so we included larger_than to have a more “algorithmic” task. If accepted, we plan to add more information about our datasets to the main body of the paper (from appendix H).
>
> Please let us know if you have any further questions!
>
> [1] https://arxiv.org/abs/2302.13971

---

> > ### Comment · Reviewer_VXXD · 2024-06-06
> >
> > Thanks for the response! I think most of the clarifications make sense. RE: calibrated few-shot, I think what I was trying to get at is slightly different because if I understood correctly, this still concerns true/false classification. I was wondering if there could be some evaluation that could tell us if preconditions for being able to determine truth/falsehood are satisfied or not (the method I mentioned might not be best instantiation of this). I might be misunderstanding something here though.
> >
> > That being said, my score was already positive to begin with, and I'm happy with keeping it positive.

---

### Decision · Program_Chairs · 2024-07-10

**Decision:**

Accept

**Comment:**

This paper reports on a series of creative interpretability analyses aimed at supporting the hypothesis that LLMs encode, in their hidden representations, their own truth-value judgments about factual claims. The experiments use behavioral techniques, probing, and interventions. The paper also introduces a new causal probe called mass-mean probing.

The general view of the reviewers (and myself) is that this is a creative, insightful paper. It is well-written, and the experiments are well-done and clearly reported. Overall, it looks like it has the potential to become an influential paper in interpretability space.